# Challenges and possibilities in offering support to family caregivers to persons with dementia of non-European background: A qualitative study

Aber Sharon Kagwa[1], Kirsi Tiitinen Mekhail[1]*, Åsa Gransjön Craftman[1], Hanne Konradsen[1,2,3], Zarina Nahar Kabir[1], Marie Tyrrell[1,4]

1 Department of Neurobiology, Care Sciences and Society, Karolinska Institutet, Huddinge, Sweden, 2 Department of Geriatrics, Hvidovre and Amager University Hospital, Hvidovre, Denmark, 3 Steno Diabetes Center Copenhagen, Copenhagen, Denmark, 4 Sophiahemmet University, Stockholm, Sweden

☯ These authors contributed equally to this work.
* kirsi.tiitinen.mekhail@ki.se

## Abstract

### Background

With population ageing and increasing ethnic diversity in Europe, it is essential to support the growing number of family caregivers (FCs) who provide daily care for persons living with dementia. In Sweden, social care professionals have a vital role in providing formal support to family caregivers within municipalities. Previous research indicates challenges for social care professionals in reaching family caregivers to persons with dementia with immigrant backgrounds. However, there is still a lack of evidence explaining why these FCs do not use available support services to the same extent as their ethnic Swedish peers. This study aimed to explore social care professionals' perspectives on challenges and possibilities in offering support to family caregivers to community-dwelling persons with dementia of non-European background.

### Methods

An explorative qualitative design was used. The data were collected through semi-structured interviews with thirteen community-based social care professionals in Sweden. The data were analysed using Systematic Text Condensation.

### Findings

Eight themes emerged from the analysis: (i) Mistrust in the system, (ii) Hard-to-Reach group, (iii) Misalignment of expectations and reality, (iv) Stigmatised situations, (v) Timely contact, (vi) Communicating with the younger generation, (vii) Promoting and adapting support, and (viii) Coordination of services.

**Data availability statement:** The datasets generated and/or analysed during the current study are not publicly available due to ethical restrictions stipulated in the ethical approval (registration number Dnr: 2023 02911 01). The approval requires that interview data respect the anonymity and integrity of participants, be accessed only as a complete dataset, and be securely stored with restricted access. For initial inquiries regarding data access procedures, the principal investigator (ZNK) may be contacted and can provide general information about the study and facilitate communication. However, all formal data access requests are handled and evaluated exclusively by a non author institutional contact in accordance with applicable ethical and legal regulations. Institutional contact for data access requests: Data Protection Officer Karolinska Institutet Email: dataskyddsombud@ki.se Phone: +46 8 524 864 73 The data are securely stored at Karolinska Institutet and are not deposited in a public repository due to the ethical and legal constraints described above.

**Funding:** This research received funding from the Swedish Council for Working Life and Social Research (GD-2023/0006). The funders had no role in study design, data collection and analysis, decision to publish, or preparation of the manuscript.

**Competing interests:** The authors have declared that no competing interests exist.

## Conclusions

The results indicate that culturally sensitive and tailor-made support can help build and maintain trust among family caregivers of non-European backgrounds and increase the uptake of offered support. Collaboration between formal and informal societal actors and community outreach through various channels, in multiple languages, can raise awareness of available support to family caregivers and increase accessibility to groups with diverse immigrant backgrounds.

## Introduction

Europe's population is ageing [1] and becoming ethnically more diverse due to an increase in global migration. About 10% of people living in the European Union (EU) are estimated to be foreign-born [2]. In Sweden, as of 2024, approximately 20% of the population consisted of immigrants and their Swedish-born descendants. Currently, the largest first-generation immigrant groups in Sweden come from Syria and Iraq followed by Finland, Poland, and Iran [3]. The term immigrant is often used inconsistently. In 1999, the Swedish government defined an immigrant as a person who migrated to Sweden and has been registered in the population register [4]. A person with an immigrant background refers to a foreign-born person who has migrated to Sweden and/or a person born in Sweden who has at least two foreign-born parents [5], (also known as a second-generation immigrant) [4].

Older age is positively associated with developing cognitive impairment and dementia. Today dementia is a global health challenge and the fourth leading cause of death in high-income countries [6]. In 2017, approximately 6.5% of Europe's overall cases of dementia were estimated to involve persons who were foreign-born [7]. Approximately 150,000 people had a dementia diagnosis in Sweden in 2019, with an anticipated increase of 62% by 2050 [8]. The National Board of Health and Welfare estimated in 2018 that about 20,000 of persons with dementia (PWD) in Sweden were foreign-born. The group of PWD including those with an immigrant background is expected to double within the next 20 years [9].

In Sweden, family caregivers (FCs) play a crucial role in providing health and social care to community-dwelling PWDs [10]. FCs are defined as those who provide unpaid care (i.e., spouses, children, friends, and relatives) [11]. Approximately 90% of PWD who have received a diagnosis in Sweden are community-dwelling [12] and a recent study reported that 24% of them do not use eldercare services (i.e., home care and senior daycare), thus relying on informal care and support [13]. Furthermore, it is common that PWD who are cohabiting, and those who are born outside of Sweden, and in particular outside of Europe, do not use eldercare services [13]. These findings are supported by a recent review reporting that FCs to PWD from immigrant backgrounds rarely use home care services for their older family members [14].

The role of informal caregivers is multifaceted, and FCs report positive experiences in their caregiving role, such as feeling a sense of purpose [15]. However, the

prolonged, complex, and progressive nature of dementia including challenges such as behavioural changes, sleeplessness, agitation, social withdrawal, and the need for the FCs to constantly be vigilant while caring for the PWD can provide financial, physical, and emotional strain for FCs. These challenges can lead to the experience of caregiver stress, which can have negative outcomes for the FC such as emotional stress, insomnia, and depressive symptoms [16].

Sweden's long-term care (LTC) system operates under a decentralised, three-tier governance model [17]. At the top national level, the government is responsible for legislation, policy development and with overview of operations of region councils and municipalities. The 21 regional councils manage healthcare services, including hospitals and outpatient care. Meanwhile, the 290 municipalities are tasked with delivering services such as home and institutional care for older adults. This decentralised structure contributes to notable differences in LTC service provision across the country [17]. An essential principle in Sweden's eldercare is the "ageing in place" policy, which promotes a person's right to remain in their own home according to their will and capacity [18]. Municipalities are obliged to provide support to FCs in their caregiving role, in accordance with the National Board of Health and Welfare, through the Social Services Act (SOSFS; 2025:400). However, most FCs in Sweden report a lack of awareness of their legal rights or whom to approach for support in social and healthcare services [19]. In Sweden, formal support for FCs is commonly provided indirectly through support aimed toward the care recipient (i.e., rehabilitation and home care). Another type of support is direct support (i.e., information and education) often provided to FCs by community-based social care professionals, either individually or in groups [20]. This support is available to anyone who self-identifies as an FC [21,22] without the necessity of a formal approval from the municipality [23]. The Social Services Act of 2009 [24] governs the assignment and work of social care professionals [23]. As a result, this professional role is relatively new in the Swedish context [25]. A recent Swedish report [25] found an average of six years of work experience (range, 0–23 years) among social care professionals. Most of the social care professionals are women [20,26] consisting of various professions including nurses, nurse assistants and social workers who have a vital role in the development of support for FCs in the municipalities [20]. The role of social care professionals is distinct from other social work professions in public institutions, primarily due to their lack of formal decision-making authority [23]. Their work includes offering counselling and providing information about available support services [20,23]. They also play a key role in educating FCs about diseases and coping strategies, as well as educating other municipal professionals (e.g., healthcare assistants) on how to provide support to FCs [20]. Many social care professionals work part-time, while, e.g., managing clinical and managerial roles within municipalities [20]. Since legislation does not clearly define the quality or scope of support for FC [21], social care professionals often lack formal job descriptions and must self-organise. As a result, assessments of FCs' needs and the support provided vary across municipalities [20,23,25].

The Swedish National Board of Health and Welfare emphasizes the importance of following up with foreign-born FCs, an area previous studies have not addressed sufficiently [27]. Previous research indicates that foreign-born FCs to PWD do not seek support to the same extent as the Swedish-born population [28]. A recent review reports that barriers for FCs to PWD of immigrant background include a lack of knowledge about dementia (e.g., its symptoms and causes) which often leads to attributing dementia symptoms to the normal processes of ageing [14]. The lack of knowledge which may lead to delayed help-seeking, has also been described by healthcare professionals from the perspective of identifying, assessing, and diagnosing dementia in older immigrants [29]. Furthermore, older immigrants living with dementia and their FCs describe a lack of culturally appropriate dementia care services [30] and how a lack of cultural sensitivity among healthcare professionals contributes to barriers to accessing support [31]. These claims are supported by a study where healthcare professionals reported having limited experience with older immigrants with dementia or cognitive impairment and their families, despite working in immigrant-dense areas [29].

As the population ages and the number of PWD grows, including those with an immigrant background, it is crucial to address the challenges faced by their FCs in daily caregiving. Social care professionals play a key role in supporting FCs in their role as informal caregivers. However, social care professionals report in previous research that FCs to PWD with immigrant backgrounds do not use available services to the same extent as their ethnic Swedish peers [23]. Furthermore,

there is still limited evidence explaining why this group of FCs does not utilise available support services. Therefore, this study aimed to explore social care professionals' perspectives on challenges and possibilities in offering support to family caregivers to community-dwelling persons with dementia of non-European background.

## Materials and methods

### Research design

A qualitative explorative design was used to gain an in-depth understanding of the participants' perspectives [32]. The 32-item COREQ (COnsolidated criteria for REporting Qualitative research) checklist was used to report this study [33].

**Study setting and recruitment** Social care professionals were eligible for inclusion if they were employed within municipal social care services in Sweden and provided support to FCs of older adults. Purposive sampling was used to identify suitable municipalities and professionals. Specifically, municipalities were selected based on having a high proportion of residents with immigrant backgrounds, as reported by the Statistics Sweden [34] to ensure relevance to the study's focus. Contact information for social care professionals (e.g., email addresses and telephone numbers) was retrieved from official municipal websites.

More than 100 email invitations were sent to social care units across the selected municipalities. As, non-responders did not actively indicate refusal, it was not possible to determine the exact number of individuals who declined to participate. Professionals who responded and expressed willingness were enrolled in the study. To broaden the pool of potential participants, snowball sampling was also employed, whereby participating professionals were invited to refer colleagues who met the inclusion criteria. The final sample size was determined based on the study's information power [35].

### Data collection

After initial contact and receiving written informed consent from participants, interviews were scheduled, based on their preferences of place of interview. The interviews were conducted primarily by the co–first authors, KTM and ASK, both of whom are registered nurses holding PhD degrees. MT, a registered nurse with a PhD, and ZNK, Associate Professor of Public Health (PhD), also participated in selected interviews between March and May 2024. Eleven interviews were held via online video calls and two were conducted in person. The length of the interviews ranged from 21.02 to 46.32 minutes. Data was collected through semi-structured interviews, guided by an interview guide (See S1 Table 1. Interview guide). The interviews included questions such as: "Have you been in contact with caregivers of non-European backgrounds? How do you reach out to them?" Following probes like "Can you provide an example?" were used to clarify responses and avoid misunderstanding. Interviews were audio recorded and transcribed verbatim by the two first authors (KTM and ASK) and cross-checked with the audio recordings.

### Data analysis

Systematic Text Condensation (STC) was employed for data analysis due to its ability to ensure intersubjectivity, reflexivity, and feasibility during the data analysis process. It is a structured, well-described, systematic method for analysing qualitative data [36]. The approach was further chosen because it provides a transparent, stepwise procedure well aligned with the descriptive aims of the study and allows for systematic cross-case thematic analysis while remaining close to participants' accounts. Furthermore, the aim was to produce a structured, practice-relevant synthesis of professionals' perspectives.

The STC consists of four steps: (i) Total impression – from chaos to themes, (ii) Identifying and sorting meaning units – from themes to codes, (iii) Condensation – from code to meaning and (iv) Synthesizing – from condensation to description and concepts [36]. Interview data were managed using Microsoft Word throughout the analysis. In the initial step of the analysis, the transcripts were read through to gain a general impression of the data, followed by a discussion between

co-authors to establish preliminary themes. Thereafter, meaning units related to the research question were identified and sorted into code groups with the initial themes in mind followed by a discussion with co-authors. Subsequently, a systematic abstraction of meaning units within the code groups established in the previous step was conducted. In the last step, the data was reconceptualized and described, including illustrative quotations. The first interview was openly discussed among co-authors, and a consensus was reached before proceeding with the analysis of the remaining interviews.

## Ethical considerations

The study received ethical approval from the Swedish Ethical Review Authority (Dnr: 2023-02911-01) and was conducted following the Declaration of Helsinki [37]. A signed informed consent was obtained from the social care professionals before the interviews. Participants were informed that their participation in the study was voluntary and that they could withdraw at any time before findings are published without providing a reason. All the data was pseudonymized and kept confidential on a secure server at Karolinska Institutet.

## Findings

The thirteen social care professionals who participated in the study were all women who provided support to FCs to PWD. The social care professionals had diverse educational backgrounds, and their current positions were based in the municipalities and included social care work, consultative positions in dementia care and coordinator/member of dementia team/ eldercare team. The social care professionals were based across Northern, Central and Southern Sweden (see Table 1). Demographic data such as participants' age or immigration status were not collected.

Eight themes emerged following the STC analysis. Five themes addressed *challenges*: (i) Mistrust in the system, (ii) Hard-to-Reach group, (iii) Misalignment of expectations and reality, (iv) Stigmatised situations, and (v) Timely contact. Three themes addressed *possibilities*: (vi) Communicating with the younger generation, (vii) Promoting and adapting support and (viii) Coordination of services.

### 1. Mistrust in the system

Social care professionals perceived mistrust in the system as a key *challenge* faced by FCs. They observed that such mistrust in the system often hindered FCs from accessing available support. This feeling was either directly experienced or subtly sensed by the professionals in their interactions with FCs, particularly regarding the Swedish healthcare system. According to the participants, this mistrust could be attributed to several factors, including FCs' previous negative experiences in their countries of origin.

Table 1. An overview of selected demographic data of the social care professionals.

| Social care professionals | | N = 13 |
|---|---|---|
| Geographical location | Northern Sweden | 3 |
| | Central Sweden | 7 |
| | Southern Sweden | 3 |
| Educational background | Specialised dementia nurse (RN) | 4 |
| | Social worker | 6 |
| | Occupational therapist | 1 |
| | Specialised enrolled nurse | 2 |
| Current job title | Social care worker at the municipality | 7 |
| | Consultative (dementia care) role in the municipality | 3 |
| | Coordinator/member of dementia team/ eldercare team at the municipality | 3 |

*"They* [the FCs to PWD of non-European backgrounds] *have other baggage with them, in everything. You know they come from such countries where you could end up in prison if you speak out … or say something against the ruling regime. "* (Social care professional 6)

Participants further stated that they perceived FCs were sometimes suspicious of them, often mistaking them for government officials. This suspicion, according to social care professionals, led FCs to believe that they were monitored or controlled through home visits. In some cases, FCs even feared that these professionals might remove the PWD from their homes against their will.

*"The FC almost felt that healthcare was an adversary, that we wanted to take the mother* [the person with dementia] *away from them. But it was not, it was for the patients' best…I have to say I thought it was a rather tough meeting.* " (Social care professional 10)

The presence of mistrust made it challenging for social care professionals to establish contact or offer support services for the FCs (i.e., support groups) and the PWD they cared for. Mistrust could stem from FCs` experiences after relocation to Sweden and stories circulating within the communities or through news outlets. These stories were often part of a distorted debate about social services in Sweden and how home care staff steal from care recipients´ homes, etc. Consequently, it was described that FCs could be reluctant to allow professionals into their homes. Social care professionals had also experienced that language barriers among the FCs could lead to mistrust in the healthcare system. For instance, a lack of understanding of the term "confidentiality" in Swedish could lead to suspicion. Participants noted that discussing private issues in support groups or requiring an interpreter could also raise suspicion among FCs. They feared that interpreters or other support group members might not maintain confidentiality, further contributing to the reluctance to engage with available support services.

*It was like a woman* [an FC] *told me at the time that –* *"I can't imagine participating in a support group because I'm not sure that the promise of confidentiality would be upheld."* (Social care professional 7)

## 2. **Hard-to-Reach group**

Participants found it *challenging* to establish contact and provide support to the FCs of PWD of non-European backgrounds. Despite multiple initiatives, they found that engaging with this group of FCs proved more difficult compared to ethnic Swedish FCs. Additionally, during community outreach efforts to offer support through local associations social care professionals noted the absence of women, who were typically the primary FCs, in these gatherings. The social care professionals further noted that the FCs to PWD of non-European backgrounds often had different support needs (i.e., monetary compensation) than those usually offered. Consequently, the contact did not often progress beyond the initial stage, where the social care professional provided information about available support services.

*"We have a skilled rehab team who often meet FCs when they need assistive devices/supportive equipment...They also reach out to those with non-European backgrounds, and I have many telephone conversations, but it does not go beyond this initial stage so much."* (Social care professional 4)

Additionally, participants also noted a lack of attendance by the FCs in support activities for informal caregivers (i.e., meetings, support groups and lectures). This was attributed to reasons such as strain related to caregiving, lack of knowledge of available support services, and caregiver's cultural norms.

*"… there are very, very few* [FCs to PWD of non-European backgrounds] *who show interest* [in support groups]. *… I can imagine that they may not have time for anything else* [than caregiving], *they just want to manage the chaos that is surrounding them…. culture, religion, language, all of this have an impact."* (Social care professional 6)

Participants also noted the FCs' preference to rely on their immediate social networks rather than municipal support. This preference could be attributed to cultural norms which emphasise taking care of one's own relatives without professional support.

3. **Misalignment of expectations and reality**

Participants noted a misalignment between expectations of the municipality and the reality faced by FCs to PWD from non-European backgrounds as a *challenge*. Social care professionals expressed that the FCs often declined support because they perceived caregiving as a duty or a way of reciprocity towards their parents. Therefore, FCs often choose to manage caregiving, independently, risking negative consequences on their physical and mental health, and financial situation.

*"A FC explained how his Mum* [the person with dementia] *had served him and now he would carry her....so no matter how much I tried to place it in a larger context, it always ended up with him feeling that he owed his mother to take care of her. That was nice but it came at the expense of the FC's well-being. I did not think I had quite the right arguments..."* (Social care professional 3)

Most social care professionals expressed that they lacked training on the diverse cultural backgrounds of the FCs they come across. This lack of knowledge may account for some of the miscommunication or misunderstanding that occurred when offering support to FCs to PWD of non-European backgrounds.

*"Just gaining more knowledge …would make it easier to reach out, if I knew what the* [FCs to PWD of non-European backgrounds] *concerns are. Knowing where to turn, what this culture provides, what the norms are, and how I can relate to them is crucial."* (Social care professional 4)

The support needs of FCs to PWD of non-European backgrounds were often perceived by the social care professionals as similar to that of their ethnic Swedish peers. However, there was a notable difference in the type of support requested and accepted by the FCs, the most common being instrumental support (i.e., aids) and financial support (i.e., monetary compensation). The social care professionals also observed that these types of support were often seen as less intrusive, making it easier for the FCs to accept, compared to emotional support or having someone else take over the caregiving of the PWD.

*"I feel that we* [social care professionals] *tell them what support is available and everything, but they are not interested in that. Instead, he* [the FC] *wanted a salary for taking care of the PWD … "* (Social care professional 8)

Social care professionals also perceived that the social network of the PWDs supported each other to a greater extent within the family system even though it was strenuous. It was further described by the social care professionals that the PWD seem to have greater expectations of their children and grandchildren to care for them at home than their ethnic Swedish peers.

4. **Stigmatised situation**

Stigma experienced by the FCs to PWD of non-European background posed a *challenge* for the social care professionals in establishing contact to offer support. According to the social care professionals, stigma could be related to FCs'

perceptions of PWD being labelled as "crazy" due to lack of knowledge about the condition. Challenging behaviours displayed by the PWD, which could be perceived as shameful for the FCs, caused some to fear speaking about their situation to others outside their close family members.

> *"It is almost shameful to talk about your mother having Alzheimer's. They* [FCs to PWD of non-European backgrounds] *do not understand the disease. That is why they do not dare to talk to others about how they experience their situation."* (Social care professional 10)

Stigma could also stem from expectations and values associated with the caregiving role within cultural communities. For instance, prioritising oneself or accepting offered support was perceived as shameful, even when the FCs lacked a social network to support them. Social care professionals also perceived that FCs could be deterred from seeking or accepting support because they feared potential shame related to their gender or negative repercussions from relatives who remained in their home countries.

> *"...in some cultures, they* [FCs to PWD of non-European backgrounds] *have pressure from those from their home countries... That you are "a bad woman," unless you take care of your or his* [partner's] *mother... many times it might not be the FCs' own choice. "* (Social care professional 12)

The social care professionals also noted that FCs could perceive accepting offered support as a sign of weakness and therefore, chose to keep to themselves.

5. **Timely contact**

Establishing timely contact posed another *challenge* for social care professionals when offering support to FCs of non-European backgrounds. The delay in contact and reluctance to accept support could result in delayed support when the FCs experienced burnout or a crisis (i.e., violence in the home), as one participant described:

> *"...we come in when there is a crisis. It would have been much easier if they had received support earlier..."* (Social care professional 1)

Therefore, the social care professionals also stressed the importance of establishing timely contact with FCs immediately after the PWD receives their diagnosis. This approach could offer the necessary support as early as possible. One social care professional suggested that they should be based in the geriatric clinics or healthcare centres where the PWD receives their diagnosis. This availability would facilitate immediate support for FCs at a critical time.

> *"When* [the person with dementia] *receives a diagnosis from the healthcare centre or geriatric clinics, a social care professional should already be there from the municipality.... [and] start to build a relationship* [with the FCs]*... "* (Social care professional 10)

Social care professionals also addressed the need for the municipalities to be more initiative-taking by improving follow-up in caregiver support after diagnosis and at various stages of dementia without the FCs having to seek the support by themselves.

6. **Communicating with the younger generation**

One of the primary methods social care professionals used as a *possibility* to establish contact and offer support for FCs to PWD of non-European backgrounds was communicating with the younger generation of the PWD (i.e., children and

grandchildren). The younger generation often did not face language barriers for communication and were, therefore, often appointed contact persons or interpreters within the family. This facilitated more effective communication and support and was viewed by participants as a possibility for improving engagement with FCs.

*"In the documentation of dementia diagnosis, it is usually contact information for the second generation* [family member] *that is stated… you can call a child and then they can sometimes say that the parents have difficulties with the language…"* (Social care professional 5)

A social care professional also noted that in addition to not having language barriers, the younger generation functioned as a point of entry to establish contact because they could identify the families' needs for help and were more open to reaching out for support.

### 7. Promoting and adapting support

Promoting and adapting support offered was another *possibility* identified by the social care professionals when establishing contact with FCs to PWD of non-European background. Some participants addressed the importance of remaining open-minded and avoiding drawing preconceived conclusions based on FCs' cultural or religious backgrounds. Instead, they advocated for meeting as fellow humans and being adaptive to the individual needs of each caregiver.

*"They* [the FCs] *are not a religion, but rather a person just like me, and what does this person need? So, it is an openness that we* [the social care professionals] *need to have."* (Social care professional 11)

However, the diversity of the multicultural backgrounds of the FCs to PWD of non-European backgrounds also needed consideration when offering support. The social care professionals noted that it was important to embrace the diversity among FCs and be attuned to their life stories and cultural traditions to provide tailor-made support. Collaborating with interpreters, learning common greeting phrases in the FCs' native language and meeting the caregivers in the comfort of their homes or through face-to-face meetings were among the approaches the social care professionals used to overcome language barriers.

"[Home visits] *have all the advantages…The people you meet feel often more secure and comfortable in their home environment…If there are language difficulties, they usually get resolved better when sitting across from each other rather than if you are on the telephone. "* (Social care professional 5)

However, the social care professionals often expressed that they could improve their communication when concerning language. Some suggestions were to clarify the meaning of "caregiving," translate their information pamphlets into different languages, and explain the benefits of the offered support in a language that the FC understands. Another suggested way for the municipalities to adapt to the needs of the FCs was to use the knowledge of culturally diverse personnel, such as home care staff who often were of non-European backgrounds.

*"Having a colleague who speaks Arabic and Somali could be a way forward. These are the most common languages in our municipality. It might also be easier to integrate into those contexts if one could convey oneself. I would see that as a great asset for this target group."* (Social care professional 4)

Social care professionals further expressed that they had adapted their support activities to better reach FCs to PWD of non-European backgrounds. They provided individual- or smaller support groups to encourage open communication. One participant also explained that her municipality had considered the FCs' varying availabilities based on their schedules

when offering support. Additionally, social care professionals working in the same municipality offered alternative support services, such as physical activities to overcome participation barriers like language.

*"We have water aerobics groups running every week, there we have participants of non-European backgrounds … we have seen this as a path to success...this with health-promoting activities, it also does not rely on conversations where language barriers become apparent."* (Social care professional 3)

The social care professional suggested reaching FCs to PWD of non-European backgrounds by offering adapted support for caregivers and the PWD such as culturally tailored support groups and senior day cares. It was noted that sharing a cultural background, such as having a physician from the same background, could help mitigate mistrust among FCs. Additionally, social care professionals realised that the most effective way to build trust and disseminate information about their support services was through word-of-mouth.

*"We know in relation to us* [in the municipality] *it is the word-of-mouth method when someone has met us and is satisfied. That is the absolute best way to advertise for us."* (Social care professional 4)

The participants emphasised the importance of providing the FCs to PWD of non-European backgrounds with a pleasant experience when offering support which could have a trickle-down effect causing others to seek or accept offered support.

## 8. Coordination of services

The social care professionals also emphasized the need to coordinate services to reach FCs to PWD of non-European backgrounds to improve the *possibility* to establish contact to offer support. One suggestion to bridge the gap between the municipalities and the FCs, was to collaborate with diverse cultural associations in immigration-dense areas. Using various outreach channels such as libraries was also recommended to enhance engagement and support for FC. Furthermore, participants discussed the possibility of co-ordinating with services that FCs regularly used (i.e., home care and primary healthcare services) within immigrant-dense communities. This approach was seen as a potential way to better reach and support FCs of non-European backgrounds.

*"… Today we're going out to speak to the primary healthcare centre…Then we can also talk about this group* [FCs to PWD of non-European backgrounds] *that they* [the healthcare centres] *should direct them* [to us] *… because they do come to the primary healthcare centres..."* (Social care professional 9)

Participants also discussed the need for municipalities to think creatively to reach FCs to PWD of non-European backgrounds by offering support services in different languages and promoting activities through alternative channels. Examples of such channels were Swedish for immigrants (SFI) classes, TV commercials and social media. A social care professional explained that she had taken a very pro-active approach to identifying and reaching this group of FCs by using citizens' guides through outreach programs:

*"…we have some outreach activities together with citizens' guides in our city...they speak different languages and are from diverse cultures. The idea has been to reach out to people with information in various languages and then speak more with them about how to consider dementia. "* (Social care professional 2)

Community engagement by using lay persons with a non-European background as ambassadors to spread the word of available support services (i.e., pupils in the schools, nurse assistant students and Swedish for Immigrants [SFI]) students were also mentioned. Furthermore, participants discussed the importance of addressing this issue at a political level to

improve outreach, develop awareness on how to address cultural diversity among healthcare professionals and incorporate it into healthcare education.

## Discussion

This current study presents new insights from social care professionals' perspectives in offering support to FCs with non-European backgrounds in Swedish municipalities. It addresses existing challenges and explores possibilities to build bridges across social, cultural and systemic divides between these FCs and social care services. Social care professionals' perceived challenges ranged from FCs mistrusting the system to holding expectations about available support that were not aligned with the offered support. This, in turn, appeared to hinder the FCs' willingness to seek support and placed the caregivers out of the social care professionals´ reach, making them a *hard-to-reach group.* Apart from identifying challenges social care professionals also proposed possibilities, including suggestions on how to improve the coordination of services and involve the younger generation in the family as facilitators. These suggestions, if implemented, could enhance offered support to FCs through strategic and organizational adaptations.

One of the main findings in this study was the reported language barriers observed among FCs of non-European backgrounds and how these barriers potentially contributed to their absence as service recipients. A combination of lacking language skills and the spread of misinformation in the community fuelled *mistrust in the system.* This mistrust created challenges for the social care professionals in establishing contact to offer support. Lacking a common language and limited awareness of available services, align with the voices of FCs caring for older migrants in other European countries as barriers to accessing dementia-related services [14,31] Furthermore, holding a negative perception of formal support services in terms of familial, cultural, and individual values can also deter FCs from seeking help [38].

Cultural barriers (i.e., religion, and customs) created through untailored support services are another integral reason for FCs to provide care themselves without accessing formal support [38]. A recent study reports the importance for older FCs of Finnish descent living in Sweden to communicate in their native language when discussing something complex (i.e., care situations) even though they could speak Swedish [39]. Services should therefore, to a certain extent, be adapted to the needs of minority ethnic groups in terms of language [40]. In this study, social care professionals described that *communicating with the younger generation* was a key strategy for addressing language barriers. As the younger generation did not face the same language barriers for communication, they were more open to receiving formal support, as also reported in a recent systematic review [14].

Building and maintaining trust in the health system among immigrants requires time [41]. Community engagement by social and healthcare professionals is therefore crucial to foster trust [42]. Religious (e.g., churches) and cultural organisations are identified as sources of informal support by FCs of immigrant backgrounds [43]. A major challenge identified by the social care professionals in this study was the misalignment of expectations of FCs and their support needs (often monetary compensation) compared to the reality of the types of support available (i.e., support groups). Previous research reports that FCs often function as a bridge between the care recipient and formal support services, filling the gap where public health services are lacking including providing personal care and cooking familiar food for the older immigrants [44]. Therefore, municipalities need to consider the importance of financial support which would compensate for the FCs' time and effort [44]. However, a recent Norwegian study of female Pakistani FCs' views on future formal and informal care for their older relatives reported contradictory findings, where the FCs expressed that accepting professional home care was considered less shameful within their communities than receiving monetary compensation for providing informal care for one's older relatives [45]. This emphasises the need for formal support providers to be aware of cultural diversities to provide tailor-made support to meet the complex needs of FCs.

Furthermore, the social care professionals in this study expressed that FCs could be reluctant to seek support despite facing a heavy care burden in terms of financial, physical- and mental strain. Research further suggests that the use of formal care services (i.e., long-term care facilities) for an older family member often contradicts the strong care norms of

FCs of immigrant background, related to their moral duties and cultures [31], such as filial piety, defined as respect for one's parents and older family members and familism which includes loyalty [46], reciprocity and strong care norms of taking care of one's own [14]. Reciprocity is integral to informal care, especially amongst FCs who have experienced the challenges of their first-generation parents (i.e., limited language skills, social network, and work conditions) [47]. The caregiving behaviours of FCs can be shaped by their collectivistic upbringing, despite Sweden being an individualistic culture [38]. These cultural influences were discussed by social care professionals who reported stigma related to potential condemnation from the FCs' community when prioritising one's own needs. Furthermore, social care professionals perceived that FCs had a greater expectation from the PWD to be taken care of through informal care than their ethnic-Swedish peers. These expectations are further addressed in a recent study where female Pakistani FCs reported that the care norms amongst their communities remained the same as in their home countries. Therefore, the first-generation immigrants expected to be supported through informal care despite migration to a foreign country and a new socio-cultural context [45]. The conditions for caring for older people at home could be more favourable in the FCs home countries due to access to support from extended family and affordable private help [45]. Taking on informal care and being of immigrant backgrounds with limited social networks can therefore lead to challenges for FCs to balance work and everyday life [14], as also perceived by the participants in this study.

The social care professionals in this study also emphasized the importance of *promoting and adapting support* as a possibility to offer support, to meet the individual needs of FCs while considering their life stories and cultural background. This study highlights a need for increased cultural awareness amongst professionals providing support services in healthcare, recognising the cultural diversity of care recipients when providing care involves respecting and acknowledging their cultural norms and backgrounds. This also entails using skilled interpreters and culturally tailored information material [42]. The social care professionals in this study similarly suggested collaborating with interpreters and translating information pamphlets (e.g., about dementia and available support services) into different languages to overcome language barriers and increase awareness among FCs of non-European backgrounds. Using the competencies of staff members of diverse cultural backgrounds (i.e., home care staff and citizens' guides) was also identified as a possibility to bridge the cultural distance and build trust. Other strategies which had been successful with FCs of immigrant backgrounds included meeting the caregivers in person and providing support through physical activity and in smaller groups. Delivering continuous training programs and effective education to health and social care professionals is crucial to overcoming cultural barriers between care providers and care recipients. These programs should focus on enhancing cultural awareness and sensitivity among professionals while considering the diverse cultural needs of immigrants [42]. Another effective approach is to integrate culturally responsive teaching methods into healthcare education (i.e., encouraging care professionals to reflect on their biases) [42].

## Strengths and limitations

All study participants were women, which reflects the gender distribution among social care professionals in Sweden [20,26]. However, including male social care professionals in the interviews could have provided a broader perspective on the topic under study. Most FCs in Sweden are women [48], a distribution also noted by the social care professionals in this study. Some FCs may have gender preferences when receiving support, potentially feeling more comfortable with professionals of the same gender. This could affect the uptake of the support offered among the FCs to PWD of non-European background.

The option provided to the participants to be interviewed in person or via online video calls contributed to overcoming practical barriers, i.e., distance which enabled wider reach to include eligible participants. Thus, social care professionals from various areas in Sweden were included, giving comprehensive data with representatives from various parts of the country. However, a limitation of the study is that demographic data for social care professionals were only partially captured, with variables such as age and immigration status omitted, which could have added valuable insight for the

analysis. The social care professionals' reflections were based on varying degrees of experience supporting FCs to PWD of non-European background. Therefore, some of the participants may have had limited experience to base their reflections on.

### Implications

This study provides valuable insights for reaching family caregivers of non-European backgrounds. It suggests that providing tailor-made and culturally sensitive support can help build trust and increase the uptake of offered support.

The findings can foster collaboration between formal and informal societal actors and promote community outreach through various channels in multiple languages. This can raise awareness of the available social support for family caregivers to persons with dementia from non-European backgrounds and make support more accessible to diverse groups with immigrant backgrounds. The current study included the perspectives of social care professionals. Exploring the topic from the perspectives of FCs to PWD of non-European background would provide important insights into the topic.

### Conclusion

The findings from this study provide valuable insights that can help community-based social care professionals adapt their support and establish contact with the hard-to-reach group of FCs of non-European backgrounds. This includes leveraging the knowledge of culturally diverse personnel and engaging with the younger generation to facilitate communication. The study suggests that culturally sensitive and tailor-made support can help build and maintain trust in the care system among FCs of non-European backgrounds and increase the uptake of offered support. Collaboration between formal and informal societal actors and community outreach through various channels, in multiple languages, can raise awareness of the available support services to FCs and make it more accessible to diverse immigrant groups.

### Supporting information

**S1 Table. This is the S1 Table Interview guide.**
(DOCX)

### Acknowledgments

The authors would like to thank the community-based social care professionals who participated in this study.

### Author contributions

**Conceptualization:** Kirsi Tiitinen Mekhail, Aber Sharon Kagwa, Åsa Gransjön Craftman, Hanne Konradsen, Zarina Nahar Kabir, Marie Tyrrell.

**Data curation:** Kirsi Tiitinen Mekhail.

**Formal analysis:** Kirsi Tiitinen Mekhail, Aber Sharon Kagwa, Åsa Gransjön Craftman, Hanne Konradsen, Zarina Nahar Kabir, Marie Tyrrell.

**Funding acquisition:** Aber Sharon Kagwa, Åsa Gransjön Craftman, Hanne Konradsen, Zarina Nahar Kabir, Marie Tyrrell.

**Investigation:** Kirsi Tiitinen Mekhail, Aber Sharon Kagwa, Zarina Nahar Kabir, Marie Tyrrell.

**Methodology:** Kirsi Tiitinen Mekhail, Aber Sharon Kagwa, Åsa Gransjön Craftman, Hanne Konradsen, Zarina Nahar Kabir, Marie Tyrrell.

**Project administration:** Kirsi Tiitinen Mekhail, Zarina Nahar Kabir.

**Resources:** Zarina Nahar Kabir.

**Supervision:** Hanne Konradsen, Zarina Nahar Kabir.

**Validation:** Kirsi Tiitinen Mekhail, Aber Sharon Kagwa, Åsa Gransjön Craftman, Hanne Konradsen, Zarina Nahar Kabir, Marie Tyrrell.

**Visualization:** Kirsi Tiitinen Mekhail, Aber Sharon Kagwa.

**Writing – original draft:** Kirsi Tiitinen Mekhail, Aber Sharon Kagwa.

**Writing – review & editing:** Kirsi Tiitinen Mekhail, Aber Sharon Kagwa, Åsa Gransjön Craftman, Hanne Konradsen, Zarina Nahar Kabir, Marie Tyrrell.

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
