## [Decision Letter · Decision Letter 0]

24 Oct 2025

PONE-D-25-46228Social care professionals' perspectives on challenges and possibilities in offering support to family caregivers to persons with dementia of non-European background: A qualitative studyPLOS ONE

Dear Dr. Tiitinen Mekhail,

Thank you for submitting your manuscript to PLOS ONE. After careful consideration, we feel that it has merit but does not fully meet PLOS ONE’s publication criteria as it currently stands. Therefore, we invite you to submit a revised version of the manuscript that addresses the points raised during the review process.

We look forward to receiving your revised manuscript.

Kind regards,

Pablo Ruisoto Palomera

Academic Editor

PLOS ONE

Journal Requirements:

This research received funding from the Swedish Council for Working Life and Social Research (GD-2023/0006).

The authors would like to thank the community-based social care professionals who participated in this study. This research received funding from the Swedish Council for Working Life and Social Research (GD-2023/0006).

This research received funding from the Swedish Council for Working Life and Social Research (GD-2023/0006).

5. In this instance it seems there may be acceptable restrictions in place that prevent the public sharing of your minimal data. However, in line with our goal of ensuring long-term data availability to all interested researchers, PLOS’ Data Policy states that authors cannot be the sole named individuals responsible for ensuring data access (http://journals.plos.org/plosone/s/data-availability#loc-acceptable-data-sharing-methods).

7. Please amend the manuscript submission data (via Edit Submission) to include author Zarina Nahar Kabir and Marie Tyrrell

8. Please amend your authorship list in your manuscript file to include author Zarina Nahr Kabir and Marie Tyrell

10. Please remove all personal information, ensure that the data shared are in accordance with participant consent, and re-upload a fully anonymized data set.

Reviewers' comments:

Reviewer's Responses to Questions

**Comments to the Author**

1. Is the manuscript technically sound, and do the data support the conclusions?

Reviewer #1: Yes

Reviewer #2: Yes

2. Has the statistical analysis been performed appropriately and rigorously? 

Reviewer #1: Yes

Reviewer #2: Yes

3. Have the authors made all data underlying the findings in their manuscript fully available?

Reviewer #1: Yes

Reviewer #2: Yes

4. Is the manuscript presented in an intelligible fashion and written in standard English?

Reviewer #1: Yes

Reviewer #2: Yes

5. Review Comments to the Author

Reviewer #1: Thank you for the opportunity to review this manuscript. Overall, the paper is well written and addresses an important and timely societal phenomenon in Europe. The study makes a valuable contribution to understanding the challenges and possibilities in supporting family caregivers of individuals with dementia from non-European backgrounds.

That said, I would like to raise the following points for consideration:

Title: The current title is rather long and somewhat dense, which may make it confusing to readers. I recommend revising it for clarity and conciseness while still capturing the essence of the study.

Abstract: The opening sentence is unclear and could be restructured for better readability and impact. A more precise formulation would strengthen the abstract.

Participants: More detail should be provided about the study participants. For instance, elaborating on their expected job roles and levels of experience would help give readers a clearer understanding of who was included in the study and the context of their perspectives.

Introduction: The introduction is well written and provides a solid foundation for the study.

Discussion: This section would benefit from more in-depth analysis of the identified themes. Expanding on the findings and critically engaging with them in light of existing literature would add significant value. Additionally, the discussion should explicitly outline recommendations for future research based on the study’s findings.

Limitations: The limitation that all participants were women should be further developed. Please explain in more detail how this may have influenced the findings and whether this reflects the broader demographics of the profession.

Participants Background: Consider adding a paragraph about the nature of social care professionals’ work, including how they are typically recruited and chosen. This background is important, as it may be directly related to the challenges and possibilities they identified.

Reviewer #2: General comments:

This study addresses an important research gap in understanding the challenges and possibilities in offering support to non-European family caregivers of persons with dementia. I believe this is significant.

6. PLOS authors have the option to publish the peer review history of their article (what does this mean?). If published, this will include your full peer review and any attached files.

Reviewer #1: No

Reviewer #2: **Yes:** Vivian Della Atuwo-Ampoh

---

## [Author Response · Author response to Decision Letter 1]

3 Nov 2025

Our respond to editor and reviewers is attached as a separate document: "Response to reviewers".

---

## [Decision Letter · Decision Letter 1]

9 Apr 2026

PONE-D-25-46228R1

Challenges and possibilities in offering support to family caregivers to persons with dementia of non-European background: A qualitative study

PLOS One

Dear Dr. Tiitinen Mekhail,

Thank you for submitting your manuscript to PLOS ONE. After careful consideration, we feel that it has merit but does not fully meet PLOS ONE’s publication criteria as it currently stands. Therefore, we invite you to submit a revised version of the manuscript that addresses the points raised during the review process.

We look forward to receiving your revised manuscript.

Kind regards,

Ilse Bloom

Staff Editor

PLOS One

Journal Requirements:

1.If the reviewer comments include a recommendation to cite specific previously published works, please review and evaluate these publications to determine whether they are relevant and should be cited. There is no requirement to cite these works unless the editor has indicated otherwise.

Additional Editor Comments:

The manuscript has been further evaluated by two reviewers, and their comments are available in the attached file/below.

Could you please carefully revise the manuscript to address all comments raised?

In addition, we request that the authors add further methodological detail on the following points:

- Sampling and participant recruitment, specifically: details about the purposive sampling; what were the inclusion criteria; numbers contacted/agreed/refused to participate; how was sample size determined;

- Duration of interviews;

- Who conducted interviews and information about their background/expertise;

- Further clarification on why this particular method (Systematic Text Condensation) was chosen for data analysis over other methods of qualitative data analysis.

Reviewer's Responses to Questions

**Comments to the Author**

1. If the authors have adequately addressed your comments raised in a previous round of review and you feel that this manuscript is now acceptable for publication, you may indicate that here to bypass the “Comments to the Author” section, enter your conflict of interest statement in the “Confidential to Editor” section, and submit your "Accept" recommendation.

Reviewer #1: All comments have been addressed

Reviewer #2: (No Response)

2. Is the manuscript technically sound, and do the data support the conclusions?

Reviewer #1: Yes

Reviewer #2: Yes

3. Has the statistical analysis been performed appropriately and rigorously?

Reviewer #1: Yes

Reviewer #2: Yes

4. Have the authors made all data underlying the findings in their manuscript fully available?

Reviewer #1: Yes

Reviewer #2: (No Response)

5. Is the manuscript presented in an intelligible fashion and written in standard English?

Reviewer #1: Yes

Reviewer #2: Yes

6. Review Comments to the Author

Reviewer #1: The authors have addressed the comments satisfactorily. While no major concerns remain, I recommend acceptance without further revision.

Reviewer #2: (No Response)

7. PLOS authors have the option to publish the peer review history of their article (what does this mean?). If published, this will include your full peer review and any attached files.

**Do you want your identity to be public for this peer review?** For information about this choice, including consent withdrawal, please see our Privacy Policy.

Reviewer #1: No

Reviewer #2: No

---

## [Author Response · Author response to Decision Letter 2]

15 Apr 2026

Dear Editor,

Thank you for considering our manuscript for publication in PLOS One and for providing constructive and valuable comments. We appreciate the time and effort devoted to reviewing our work.

We have carefully addressed all points raised. The revisions are highlighted in the document titled “Revised Manuscript with Track Changes” and are also summarized in the accompanying response table.

We hope that these revisions meet the journal’s expectations and enhance the clarity and quality of the manuscript. We remain grateful for the opportunity to revise and resubmit our work.

Sincerely,

Kirsi Tiitinen Mekhail

Corresponding author

Comments Action taken

Reviewer #2's comment #7

See the main reviewer comment: (Reviewer #2's comment #7)

“Page 17, line 201, Table 1 presents demographic data on social care professionals. Were the ages of these professionals recorded? Age could be an influencing factor, as older professionals might bring more life experience compared to younger ones, have developed coping strategies, and have a better understanding of the challenges. Additionally, it's worth noting the immigration status of the social care professionals themselves. Were any of them immigrants? This could impact their perspectives, potential challenges and enhance the possibilities in offering support to family caregivers to persons with dementia.”

We thank the editor for drawing attention to Reviewer 2’s comment#7.

In response, we have revised the manuscript by adding description that:

“Demographic data such as participants’ age or immigration status were not collected.” (page 11 line 245).

Additionally, we have added a statement in the limitations section acknowledging the partial collection of demographic data and the potential implications this may have for the interpretation of the findings.

(Page 27 line 626): “However, a limitation of the study is that demographic data for social care professionals were only partially captured, with variables such as age and immigration status omitted, which could have added valuable insight for the analysis. “

Editor’s comment:

- Sampling and participant recruitment, specifically: details about the purposive sampling; what were the inclusion criteria; numbers contacted/agreed/refused to participate; how was sample size determined...

We thank the editor for raising this point. We have revised Study settings and recruitment section to address the raised comment: (page 8 line 174-)

Study setting and recruitment

“Social care professionals were eligible for inclusion if they were employed within municipal social care services in Sweden and provided support to FCs of older adults. Purposive sampling was used to identify suitable municipalities and professionals. Specifically, municipalities were selected based on having a high proportion of residents with immigrant backgrounds, as reported by the Statistics Sweden [34] to ensure relevance to the study’s focus. Contact information for social care professionals (e.g., email addresses and telephone numbers) was retrieved from official municipal websites.

More than 100 email invitations were sent to social care units across the selected municipalities. As, non responders did not actively indicate refusal, it was not possible to determine the exact number of individuals who declined to participate. Professionals who responded and expressed willingness were enrolled in the study. To broaden the pool of potential participants, snowball sampling was also employed, whereby participating professionals were invited to refer colleagues who met the inclusion criteria. The final sample size was determined based on the study’s information power...”

Editors comment: - Duration of interviews The duration of the interviews has been added (Page 9 line 202-):

“The length of the interviews ranged from 21.02 to 46.32 minutes.”

Editors comment:

- Who conducted interviews and information about their background/expertise; We thank the editor for this question. Clarification regarding who conducted the interviews has been added under Data collection: (page 9 line 199):

“… conducted primarily by the co–first authors, KTM and ASK, both of whom are registered nurses holding PhD degrees. MT, a registered nurse with a PhD, and ZNK, Associate Professor of Public Health (PhD), also participated in selected interviews...”

Editor’s comment: - Further clarification on why this particular method (Systematic Text Condensation) was chosen for data analysis over other methods of qualitative data analysis.

We thank the Editor for this comment. We have addressed this point by adding the following sentences under Data analyses (page 10 line 214-):

“The approach was further chosen because it provides a transparent, stepwise procedure well aligned with the descriptive aims of the study and allows for systematic cross case thematic analysis while remaining close to participants’ accounts. Furthermore, the aim was to produce a structured, practice relevant synthesis of professionals’ perspectives.”

---

## [Decision Letter · Decision Letter 2]

27 Apr 2026

Challenges and possibilities in offering support to family caregivers to persons with dementia of non-European background: A qualitative study

PONE-D-25-46228R2

Dear Dr. Tiitinen Mekhail,

We’re pleased to inform you that your manuscript has been judged scientifically suitable for publication and will be formally accepted for publication once it meets all outstanding technical requirements.

Kind regards,

Sreeram V. Ramagopalan

Academic Editor

PLOS One

Additional Editor Comments (optional):

Reviewers' comments:

Reviewer's Responses to Questions

**Comments to the Author**

1. If the authors have adequately addressed your comments raised in a previous round of review and you feel that this manuscript is now acceptable for publication, you may indicate that here to bypass the “Comments to the Author” section, enter your conflict of interest statement in the “Confidential to Editor” section, and submit your "Accept" recommendation.

Reviewer #1: All comments have been addressed

Reviewer #2: (No Response)

2. Is the manuscript technically sound, and do the data support the conclusions?

Reviewer #1: Yes

Reviewer #2: (No Response)

3. Has the statistical analysis been performed appropriately and rigorously? 

Reviewer #1: Yes

Reviewer #2: (No Response)

4. Have the authors made all data underlying the findings in their manuscript fully available?

Reviewer #1: Yes

Reviewer #2: (No Response)

5. Is the manuscript presented in an intelligible fashion and written in standard English?

Reviewer #1: Yes

Reviewer #2: (No Response)

6. Review Comments to the Author

Reviewer #1: All comments raised in the original submission have been addressed. In this revision, the comments provided by the editor have also been incorporated and addressed accordingly.

Reviewer #2: I have re-evaluated the revised manuscript following the authors’ response to the initial round of review. The authors have addressed the comments and suggestions raised. The revisions are integrated into the manuscript. I believe this work addresses an important and timely issue. The authors appropriately acknowledge most of the remaining concerns raised as limitations of the study.

I recommend the manuscript is accepted.

7. PLOS authors have the option to publish the peer review history of their article (what does this mean?). If published, this will include your full peer review and any attached files.

Reviewer #1: No

Reviewer #2: No

---

## [Editor Report · Acceptance letter]

PONE-D-25-46228R2

PLOS One

Dear Dr. Tiitinen Mekhail,

I'm pleased to inform you that your manuscript has been deemed suitable for publication in PLOS One. Congratulations! Your manuscript is now being handed over to our production team.

Kind regards,

on behalf of

Dr. Sreeram V. Ramagopalan

Academic Editor

PLOS One